# An Integrated Medical-Psychological Approach in the Routine Care of Patients with Type 2 Diabetes: A Pilot Study to Explore the Clinical and Economic Sustainability of the Healthcare Intervention

Mara Lastretti [1], Manuela Tomai [2], Natalia Visalli [3], Francesco Chiaramonte [3], Renata Tambelli [2] and Marco Lauriola [4,*]

1   Osservatorio di Psicologia in Cronicità, Ordine Psicologi Lazio, Via del Conservatorio 91, 00186 Rome, Italy; mara.lastretti@gmail.com
2   Department of Dynamic, Clinical Psychology and Health Studies, "Sapienza" University Rome, Via degli Apuli 1, 00185 Rome, Italy; tomai.manuela@uniroma1.it (M.T.); renata.tambelli@uniroma1.it (R.T.)
3   UOC Dietologia e Diabetologia ASL Roma1 V.le Angelico 28, 00195 Rome, Italy; visallitoto@gmail.com (N.V.); franc.chiaramonte@yahoo.it (F.C.)
4   Department of Psychology of Developmental and Socialisation Processes, "Sapienza" University Rome, Via dei Marsi 78, 00185 Rome, Italy
*   Correspondence: marco.lauriola@uniroma1.it

**Abstract:** The economic burden of Type 2 Diabetes Mellitus (T2DM) is a challenge for sustainability. Psychological factors, healthy behaviors, and stressful conditions are predictive and prognostic factors for T2DM. Focusing on psychological factors can reduce costs and help ensure the sustainability of diabetes care. The study aimed to support an integrated medical-psychological approach in the care of patients with T2DM. A group of patients undergoing usual healthcare treatment was compared to patients who received a psychotherapeutic intervention in addition to standard treatment. The study's outcomes were: physical health (blood glucose, glycated hemoglobin, blood lipids, blood pressure); lifestyle (cigarettes, alcoholic drinks, physical activity, body mass index); mental health (anxiety, depression, stress, coping styles, alexithymia, emotion regulation, locus of control); costs (number of referrals to a specialist, standard cost of each visit). We examined the change from baseline to 24-week follow-up. Compared to the Standard Group, the Integrated Group reported a reduction in blood lipids and triglycerides, chronic depressive and anxious mood states, patient emotional coping, and the number of specialist visits and diagnostic tests. Close collaboration between diabetologists and psychologists is feasible, and it is worth considering integrated care as an option to contain and make healthcare spending more sustainable.

**Keywords:** Diabetes Mellitus; Type 2; sustainability of healthcare interventions; integrated treatment; multidisciplinary team; psychotherapeutic intervention

## 1. Introduction

The spread of chronic diseases is a predominant problem and challenge for global health and the sustainability of healthcare [1]. In particular, diabetes represents an epidemic condition with a high impact on the costs of national health systems worldwide [2]. In European countries, including Italy, the cost of treating diabetic patients exceeds 8% of total healthcare expenditure [3]. The diabetic population has an annual health expenditure 3–4 times higher than the rest of the population, both medical and healthy. In people with diabetes, the treatment of complications increases the costs for hospitalization (about 50% of the total), medication (about 25%), and outpatient services (diagnostics and visits, about 20%) [4,5]. People with diabetes visit physicians' offices more frequently than people without diabetes [6] due to the many complications of diabetes treatment [7]. The expected

increase in the prevalence of diabetes endangers the clinical and economic sustainability of demand for healthcare.

Sustainability research can significantly improve the impact of public health initiatives by identifying variables and procedures that maximize the intervention's efficacy and allow it to continue. What factors determine the efficacy and sustainability of interventions? Along with genetic predisposition, several individual factors such as mindless eating and sedentary can influence disease onset and progression [8]. Consequently, effective diabetes management depends on cognitive, relational, and social factors influenced by individual behaviors. There is increasing evidence that psychological factors (e.g., depression, anxiety, alexithymia), healthy behaviors (e.g., physical activity, not smoking, healthy diet), and enduring stress conditions are predictive and prognostic factors for T2DM [9–11].

Improving the quality of diabetes treatment by focusing on psychological factors can reduce costs [12,13] and help ensure diabetes care sustainability.

### 1.1. Psychological Conditions and Stress as Modifiable Risk Factors

Depression and anxiety are the most prevalent psychological comorbidities in diabetes [14,15]. Meta-analytic evidence [16,17] and longitudinal studies [18,19] agree that poor mental health increases the risk of developing more severe T2DM.

In particular, depression and anxiety were found to be associated with worse glycemic control [20,21], lower adherence to treatment [22,23], and, consequently, long-term complications and higher healthcare costs [24,25].

Psychological stress is also cited among the environmental factors that contribute to the risk of developing T2DM [9]. According to [26], enduring distress can have biological consequences, dysregulating several physiological mechanisms aimed at restoring the body homeostasis (e.g., hypothalamic–pituitary–adrenocortical axis, the autonomic nervous system, and the metabolic and immune system) [27]. Allostatic load reflects the cumulative burden of chronic stress, and its impact is assessed using several metabolic, inflammatory, neuroendocrine, and cardiovascular biomarkers [28].

Recent studies have shown that, compared to healthy controls, patients with T2DB have a significantly higher allostatic load [29,30]. Emotional burden, worry, and helplessness resulting from the daily management of the disease [31] can increase glucose levels in T2DM patients due to a generalized dysregulation of metabolic, inflammatory, cardiovascular, and neuroendocrine functions [30,32].

### 1.2. Integrated Interventions Strategies for Persons with Type 2 Diabetes

The incidence and severity of T2DM have long stimulated research and implementation of clinical intervention programs. Despite the vast heterogeneity of existing models, there is strong evidence to support the effectiveness of psychological programs in improving symptoms of depression and anxiety in patients with T2DM, reducing psychological distress, and improving glycemic control [15,33].

Patients with complex health needs, such as T2DM, can benefit from integrated care models [34]. According to Castelnovo and colleagues [35], clinical psychology and medicine must work jointly to address the challenge of healthcare sustainability. Close collaboration between diabetologists and psychologists has long been promoted [36,37], as the multidisciplinary approach is proven to be a best practice to prevent or limit complications and is widely endorsed by the diabetes community [34,38].

The small proportion of persons with diabetes who receive psychological care in public settings, as in Italy [39], suggests that more research comparing integrated and routine treatment is needed [13]. Including an economic component in this research is strongly recommended and should be the default approach [40] to support the economic sustainability of psychology services and aid policymakers in their evaluation [35,40].

Accordingly, the present study's general aim was to collect preliminary evidence supporting the clinical and economic effectiveness of integrated medical treatment and psychological intervention in the care of patients with T2DM.

We invited patients treated at a public diabetes center to participate in a treatment program that included group psychotherapy sessions in addition to the standard biomedical treatment. For comparison, we selected a second group of patients from the same center, matched in gender, age, and educational level. The study outcomes were medical, behavioral (lifestyle), psychological, and economic. The overall hypotheses of the study were that patients taking the integrated treatment had better health outcomes.

## 2. Materials and Methods

### 2.1. Study Design and Procedures

The funding institution (Ordine degli Psicologi del Lazio) aimed to collect clinical and economic evidence to support the use of an integrated medical-psychological approach in routine care of T2DM patients. The study was designed as non-randomized outcome research (Figure 1). A group of patients (Standard Treatment) undergoing the usual health treatment was compared to a group of patients who received a psychotherapeutic group intervention in addition to the standard treatment (Integrated Treatment). Approval of the Department of Clinical Dynamic Psychology and Health Study (Sapienza-Roma, Italy) ethics committee was obtained. The research complied with the Helsinki Convention norms and its subsequent amendments.

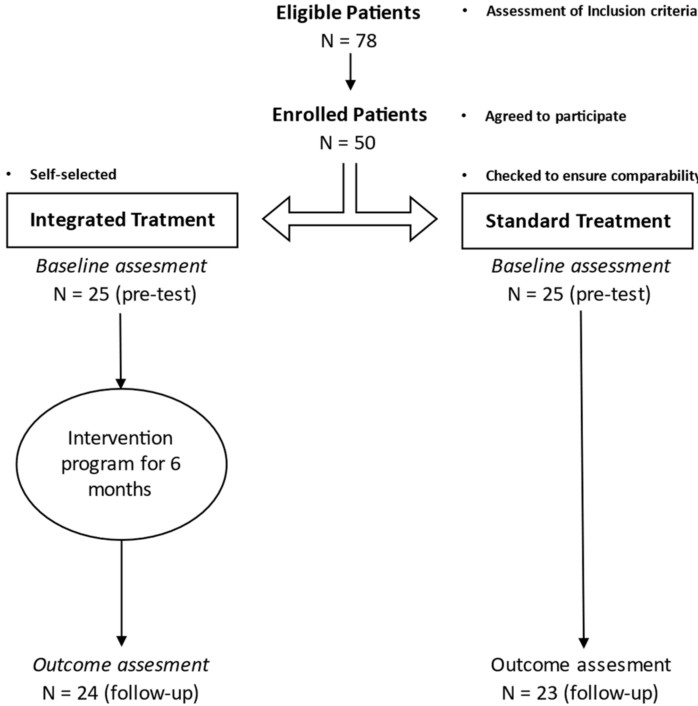

**Figure 1.** Study protocol flow chart.

Individuals over 18 years with T2DM, without a certified psychiatric disorder, having started a drug therapy (oral or multi-injective) no longer than two months before the start of the study, were recruited at the Dietology and Diabetology clinic of ASL RM1 in Rome. We considered the transition to drug therapy to be a specific critical period that could put people in a condition of greater equality, regardless of the seniority of the diagnosis. A physician identified eligible patients at the Diabetes Center. This method was chosen to increase participation rates and to ensure that the recruited sample was as close to the clinical center's territorial population. Eligible patients were sent an invitation letter describing the study's general purpose and inviting them to participate. A total of 78 patients were contacted, and 50 agreed to participate. Each patient was given a complete description of the study, including information about voluntary assignment to a group receiving integrated care. The integrated care group was self-selected. The Standard

Treatment (ST) group was selected from all patients treated at the Diabetes Center and checked to ensure comparability at baseline. Each group included 25 patients with DMT2. Confidentiality was assured to all patients regarding all data collected during the study, and informed consent was collected. No patients refused to participate in the study.

### 2.2. Participants

A total of 50 patients with DTM2 (21 Females, 51.2%) aged between 41 and 80 years old (M = 59.22 years, SD = 9.74) participated in the study. The average years of educations were 15 (SD = 3.01), and 91.9% of the sample had at least a high-school degree. Detailed descriptive statistics for study outcomes are shown in Table 1.

**Table 1.** Descriptive statistics of participants.

|  | N | M | SD | Min | Max | Range |
|---|---|---|---|---|---|---|
| **Physical Health** |  |  |  |  |  |  |
| hBa1C (%) | 47 | 8.33 | 1.66 | 5 | 13 | 7 |
| Blood glucose (mg/dL) | 47 | 175.51 | 72.79 | 87 | 350 | 263 |
| Total Cholesterol (mg/dL) | 47 | 183.57 | 36.09 | 90 | 284 | 194 |
| HDL Cholesterol (mg/dL) | 47 | 103.94 | 28.89 | 32 | 100 | 68 |
| LDL Cholesterol (mg/dL) | 47 | 59.40 | 10.46 | 48 | 208 | 160 |
| Triglycerides (mg/dL) | 43 | 110.34 | 45.61 | 32 | 226 | 194 |
| Systolic Blood Pressure (mmHg) | 47 | 119.88 | 13.35 | 90 | 170 | 80 |
| Diastolic Blood Pressure (mmHg) | 47 | 75.31 | 9.02 | 60 | 100 | 40 |
| **Lifestyle** |  |  |  |  |  |  |
| Cigarettes per day (number) | 47 | 7.98 | 11.71 | 0 | 40 | 40 |
| Wine per day (glasses) | 37 | 0.78 | 1.27 | 0 | 5 | 5 |
| Alcohol per day (glasses) | 38 | 0.34 | 0.78 | 0 | 3 | 3 |
| Walk per day (minutes) | 47 | 25.76 | 31.30 | 0 | 120 | 120 |
| Physical activity per week (minutes) | 45 | 62.18 | 86.39 | 0 | 480 | 480 |
| BMI (kg/m$^2$) | 42 | 30.84 | 593.00 | 20 | 46 | 26 |
| **Mental Health** |  |  |  |  |  |  |
| Depression | 41 | 11.43 | 8.87 | 0 | 37 | 37 |
| State Anxiety | 41 | 42.26 | 11.74 | 20 | 65 | 45 |
| Trait Anxiety | 41 | 44.42 | 10.40 | 23 | 65 | 42 |
| Health Locus of Control | 41 | 35.86 | 6.28 | 15 | 50 | 35 |
| Task-oriented Coping | 41 | 52.27 | 15.35 | 27 | 80 | 53 |
| Emotion-oriented Coping | 41 | 40.53 | 9.51 | 17 | 60 | 43 |
| Avoidance-oriented Coping | 41 | 41.09 | 9.61 | 24 | 61 | 37 |
| Difficulty Identifying Feelings | 41 | 15.77 | 5.42 | 7 | 24 | 17 |
| Difficulty Describing Feelings | 41 | 12.68 | 4.97 | 5 | 23 | 18 |
| Externally-oriented Thinking | 41 | 19.30 | 4.57 | 8 | 26 | 18 |
| Alexithymia | 41 | 48.02 | 12.13 | 20 | 68 | 48 |
| Cognitive Reappraisal | 37 | 4.75 | 1.18 | 2 | 7 | 5 |
| Suppression | 37 | 3.86 | 1.44 | 1 | 7 | 6 |

Note: N is the number of non-missing values.

### 2.3. Interventions

The ST was performed by a multidisciplinary team, including a diabetologist, a diabetes specialist nurse, and a dietician. The diabetologist monitored the patient's physical health during the study and adjusted the drug therapy accordingly; the nurse provided health education; the dietician taught carbohydrate counting.

The Integrated Treatment (IT) added participation in a group psychotherapeutic intervention to ST. In this context, we intend "psychotherapeutic" as an intervention using the therapeutic alliance between the patient and the therapist to promote emotional, cognitive, or behavioral changes that support people in changing their lifestyle and improving their chronic condition [41]. We used a group intervention approach because it is more cost-effective and appropriate in dealing with patient needs in the medical setting [42,43].

The psychotherapeutic intervention was scheduled in 12 bi-weekly group meetings lasting 60 min over six months.

The treatment model adopted refers to some of the most commonly used treatment models for the management of T2DM: Gestalt therapy [44] and Cognitive Behavioral Therapy (CBT) [45]. The program has sequentially activated emotional, relational, and cognitive processes to help participants to increase awareness of emotions related to the condition of a person with diabetes, improve interpersonal relationships or change their expectations about them, lower levels of anxiety or deflected mood, and foster the creation of a support network in the group. Attention to the present (i.e., here and now) and the experiential (i.e., how) dimensions of the self-knowledge and change process were borrowed from both treatment models. The phenomenological method develops the ability to stay in touch with the experiential dimension, keeping out as much as possible beliefs, values, theories, interpretations, previous knowledge, reflecting the Gestalt treatment model more specifically. The "empty chair technique" and "dream reprocessing" are two Gestalt therapy techniques used to help patients increase their awareness of immediate experiences, identify their feelings, and gain insight into their relationships [46]. The "cycle of contact" [47] has encouraged patients to start the process of "self-regulation of the organism", and the frequent use of metaphorical and narrative techniques (e.g., self-presentation) has facilitated the exploration of their emotions and the different parts of themselves, including the sick self and the experience of the disease. The ABC approach [48] and behavior modeling [49] are CBT techniques widely used in the proposed treatment. Using ABC, the therapist guided group participants to search for negative, automatic, maladaptive thoughts, encouraging a cognitive restructuring of the severity of the disease and a sense of greater control. The modelling technique had great use in the IT group. The group setting offered participants an opportunity to experience multiple models from which they could learn strategies for coping with the burden of illness. More generally, the group setting also played, at the same time, functions of emotional support and containment.

The program was divided into three phases: The first phase (2 weeks) was dedicated to knowing the course participants to create a climate conducive to change.

The second phase (5 weeks) dealt with analyzing the weaknesses of the patient's condition, the relationship with their caregivers, and the visits with the team. The third phase (5 weeks) dealt with elaborating feelings of helplessness and frustration and anxiety states.

To control undesired variability associated with the healthcare team or psychologists' caring style, the same medical team and psychotherapist (with certified training and more than five years of practice) treated patients.

### 2.4. Outcome Measures at Baseline and 6-Month Follow-Up

We collected the following measures:

Physical Health Outcomes; Data from medical records. The study's primary endpoints were blood glucose and glycated hemoglobin (HbA1c). Secondary endpoints were the markers of allostatic loads: blood lipids (total, HDL, and LDL cholesterol, triglycerides), and blood pressure (systolic and diastolic).

Lifestyle Outcomes. Information from patients, namely cigarettes per day, wine glasses per day, other alcoholic beverages per day, walked minutes per day, physical activity hours per week, and the body mass index (BMI), was obtained from the medical record.

Mental Health Outcomes. We invited research participants to complete a set of psychometric scales. The study's primary endpoints were depression (Beck Depression Inventory) [50] and anxiety levels (State–Trait Anxiety Inventory) [51]. Secondary endpoints were: coping styles (Coping Styles Inventory [52], alexithymia (Toronto Alexithymia Scale) [53], rmotion regulation (Emotion Regulation Questionnaire [54]), and health locus of control (Health Locus of Control Scale [55]).

Economic Outcomes. In Italy, all citizens are covered by public health insurance; therefore, each specialistic consultation might reveal a potential worsening in a patient's health and is an additional cost borne by the national health system. The primary economic

outcomes were the number of referrals to a specialist (e.g., diabetologist, ophthalmologist, and cardiologist) and each visit's standard cost. We also assessed the number of visits to the general practitioner, the number of blood tests, and diagnostic exams. All these outcomes were evaluated cumulatively during the study period and used as a proxy for health care's economic impact.

### 2.5. Statistical Analysis

Psychosocial interventions effectively reduced anxiety and depression symptoms in T2DM patients with a considerable effect size (i.e., SMD = 1.50) and improved glycemic control, with SMD-s ranging from 0.81 to 1.15 for fasting blood glucose and glycated hemoglobin, respectively (Xie and Deng, 2017) [15]. We conducted an a priori power analysis to determine the sample size needed for our study using the "pwr" package for R [56]. Assuming the effect sizes mentioned above as our best guess for the expected differences between IT and ST at follow-up, a sample of 25 participants in each treatment group would detect a difference in fasting blood glucose with alpha level 0.05 and power 0.80. For glycated hemoglobin, a sample of 13 participants in each treatment group was be needed. Given the higher effect size for mental health outcomes, the same analysis recommended sample size of 8 patients in each group. A sample size of 25 would detect a significant difference in anxiety and depression, with a power of 0.999.

We used standard inferential analyses (independent sample T-test and Chi-square test) to compare the baseline and follow-up groups. A series of ANCOVA-s investigated group differences at follow-up, adjusting for baseline values. The study hypotheses concerning longitudinal change were tested using a linear mixed model analysis for each outcome. Design factors were treatment type (Standard vs. Integrated), time (Baseline vs. Follow up), and their interaction (treatment x time). Along with the fixed effects, we specified a random intercept in the analysis to control for nonindependence of patient data, because each patient might have systematically correlated outcomes across measurement occasions. The "lme4" package for R [57] was used for model fitting with restricted maximum likelihood estimators (REML).

## 3. Results

### 3.1. Patients Characteristics at Baseline

There were no differences between groups regarding gender ($\chi^2 = 0.59$, df = 1, $p = 0.442$), age ($t = 0.51$; df = 39; $p = 0.612$), and educational level ($t = 0.33$; df = 39; $p = 0.744$). Table 2 reports detailed descriptive statistics for study outcomes at baseline, broken down by group. Regarding the primary endpoints, there were no significant differences at baseline in hBa1C and Blood glucose. The allostatic load was comparable between ST and IT. The IT group tended to have higher total and LDL cholesterol levels and lower blood pressure, but none of these differences were significant. Triglyceride levels were the same between groups at baseline.

Regarding lifestyle outcomes, wine intake per day was slightly higher for the IT group than the ST group. Hours of physical activities per week and minutes walked per day were indistinguishable between groups. IT group tended to have a higher BMI. Notwithstanding these tendencies, no lifestyle variable was significantly different at baseline. Concerning psychological outcomes, the two groups started the study with similar levels of depression. Unfortunately, the ST group reported significantly higher trait and state anxiety. Considering the other psychological outcomes, the ST group tended to report higher health locus of control, suppression of emotions, and alexithymia (especially the externally-oriented thinking) than patients in the IT condition. However, only the externally-oriented thinking score approached the conventional levels of statistical significance.

**Table 2.** Patients Characteristics at Baseline.

| | | Standard Treatment | | Integrated Treatment | | | | | |
|---|---|---|---|---|---|---|---|---|---|
| **Physical Health** | **N** | **M** | **SD** | **M** | **SD** | ***t*-Test** | **df** | ***p*** | **Cohen's d** |
| hBa1C (%) | 47 | 7.98 | (1.46) | 8.64 | (1.80) | −1.37 | 45 | 0.179 | 0.40 |
| Blood glucose (mg/dL) | 47 | 176.68 | (76.38) | 174.48 | (71.05) | 0.1 | 45 | 0.919 | 0.03 |
| Total Cholesterol (mg/dL) | 47 | 174.68 | (33.04) | 191.4 | (37.49) | −1.61 | 45 | 0.114 | 0.47 |
| HDL Cholesterol (mg/dL) | 47 | 96.9 | (19.84) | 110.14 | (34.21) | −1.59 | 45 | 0.118 | 0.47 |
| LDL Cholesterol (mg/dL) | 47 | 60.62 | (12.91) | 58.33 | (15.00) | 0.56 | 45 | 0.581 | 0.16 |
| Triglycerides (mg/dL) | 43 | 110.5 | (33.47) | 110.22 | (53.35) | 0.02 | 41 | 0.984 | 0.01 |
| Systolic Blood Pressure (mmHg) | 47 | 121.78 | (12.73) | 118.2 | (13.91) | 0.92 | 44 | 0.364 | 0.27 |
| Diastolic Blood Pressure (mmHg) | 47 | 77.03 | (10.13) | 73.8 | (7.81) | 1.23 | 45 | 0.224 | 0.36 |
| **Lifestyle** | | | | | | | | | |
| Cigarettes per day (number) | 47 | 10.14 | (13.50) | 6.08 | (9.76) | 1.19 | 45 | 0.240 | 0.35 |
| Wine per day (glasses) | 37 | 0.33 | (0.72) | 1.09 | (1.48) | −1.84 | 35 | 0.075 | 0.61 |
| Alcohol per day (glasses) | 38 | 0.07 | (0.27) | 0.13 | (0.61) | −0.31 | 36 | 0.759 | 0.10 |
| Walk per day (minutes) | 47 | 20.45 | (28.36) | 30.42 | (33.55) | −1.09 | 45 | 0.281 | −0.32 |
| Physical activity per week (minutes) | 45 | 72.14 | (107.46) | 53.47 | (63.77) | 0.72 | 43 | 0.476 | 0.22 |
| BMI (kg/m$^2$) | 42 | 23.23 | (5.04) | 29.50 | (6.41) | −1.33 | 40 | 0.191 | 0.41 |
| **Mental Health** | | | | | | | | | |
| Depression | 41 | 14.14 | (9.51) | 9.3 | (7.90) | 1.78 | 39 | 0.083 | 0.56 |
| State Anxiety | 41 | 46.95 | (9.54) | 38.58 | (12.16) | 2.4 | 39 | 0.021 | 0.75 |
| Trait Anxiety | 41 | 49.02 | (7.93) | 40.83 | (10.83) | 2.69 | 39 | 0.010 | 0.85 |
| Health Locus of Control | 41 | 37.68 | (6.00) | 34.43 | (6.24) | 1.68 | 39 | 0.100 | 0.53 |
| Task-oriented Coping | 41 | 49.79 | (16.06) | 54.22 | (14.83) | −0.92 | 39 | 0.365 | 0.29 |
| Emotion-oriented Coping | 41 | 39.43 | (8.99) | 41.39 | (10.00) | −0.65 | 39 | 0.519 | 0.20 |
| Avoidance-oriented Coping | 41 | 38.71 | (9.40) | 42.96 | (9.57) | −1.42 | 39 | 0.163 | 0.45 |
| Difficulty Identifying Feelings | 41 | 16.47 | (5.60) | 15.23 | (5.34) | 0.72 | 39 | 0.475 | 0.23 |
| Difficulty Describing Feelings | 41 | 13.62 | (4.44) | 11.95 | (5.33) | 1.06 | 39 | 0.294 | 0.34 |
| Externally-oriented Thinking | 41 | 20.73 | (3.95) | 18.18 | (4.79) | 1.82 | 39 | 0.076 | 0.57 |
| Alexithymia | 41 | 51.42 | (11.30) | 45.36 | (12.33) | 1.62 | 39 | 0.114 | 0.51 |
| Cognitive Reappraisal | 37 | 4.7 | (1.29) | 4.78 | (1.14) | −0.18 | 35 | 0.859 | 0.06 |
| Suppression | 37 | 4.29 | (1.40) | 3.61 | (1.43) | 1.41 | 35 | 0.168 | 0.48 |

Note: N is the number of non-missing values.

### 3.2. Patients Characteristics at Follow-Up

Table 3 reports detailed descriptive statistics for study outcomes at follow-up, broken down by group. After six months of treatment, there were still no significant differences in hBa1C and blood glucose. While the allostatic load was overall comparable between groups at baseline, the IT group had significantly higher HDL cholesterol and lower Triglycerides than the ST at follow-up. Significant differences in blood pressure, not significant at baseline, were different at follow-up: the IT group had significantly lower systolic blood pressure than patients in the ST group, while the diastolic one was marginally significant. No differences emerged at follow-up regarding lifestyle outcomes, except a marginally significant one in BMI. The ST group had a higher BMI than the IT group. In the psychological outcomes, the IT group had significantly lower depression scores at follow-up, where the two groups were not significantly different at the beginning of the study. Likewise, the gap in trait– and state–anxiety was much larger at follow-up than baseline. While there were no differences between treatments in coping scales at the beginning of the study, patients in the IT group had higher situational and lower emotional coping than the ST at follow-up. The alexithymia score of the two groups was statistically significant at follow-up, with the IT group being lower than the ST. Externally-oriented Thinking style was lower in the IT than in the ST, as found at the beginning of the study. Unlike baseline, the IT group had lower Difficulty Identifying Feelings than patients in the ST.

**Table 3.** Patient Characteristics at Follow-up.

| Physical Health | N | Standard Treatment | | Integrated Treatment | | t-Test | df | p | Cohen's d |
|---|---|---|---|---|---|---|---|---|---|
| | | M | SD | M | SD | | | | |
| hBa1C (%) | 45 | 6.88 | (0.91) | 7.11 | (0.96) | −0.8 | 43 | 0.426 | 0.24 |
| Blood glucose (mg/dL) | 47 | 124.76 | (30.97) | 116.31 | (20.86) | 1.11 | 45 | 0.273 | 0.32 |
| Total Cholesterol (mg/dL) | 47 | 160.95 | (26.05) | 176.99 | (22.72) | −2.25 | 45 | 0.029 | 0.66 |
| HDL Cholesterol (mg/dL) | 46 | 87.14 | (19.00) | 94.01 | (22.18) | −1.12 | 44 | 0.270 | 0.33 |
| LDL Cholesterol (mg/dL) | 46 | 52.33 | (13.66) | 61.54 | (15.36) | −2.13 | 44 | 0.039 | 0.63 |
| Triglycerides (mg/dL) | 43 | 136.44 | (54.65) | 97.2 | (46.23) | 2.54 | 41 | 0.015 | 0.79 |
| Systolic Blood Pressure (mmHg) | 46 | 120.32 | (9.62) | 114.97 | (6.22) | 2.27 | 44 | 0.028 | 0.67 |
| Diastolic Blood Pressure (mmHg) | 47 | 73.45 | (7.98) | 69.6 | (6.76) | 1.79 | 45 | 0.080 | 0.52 |
| **Lifestyle** | | | | | | | | | |
| Cigarettes per day (number) | 47 | 4.41 | (6.99) | 2.21 | (4.12) | 1.34 | 45 | 0.188 | 0.39 |
| Wine per day (glasses) | 38 | 0.20 | (0.56) | 0.26 | (0.54) | −0.33 | 36 | 0.740 | 0.11 |
| Alcohol per day (glasses) | 38 | 0.07 | (0.27) | 1.25 | (6.12) | −0.72 | 36 | 0.479 | 0.24 |
| Walk per day (minutes) | 47 | 40.5 | (31.89) | 61.93 | (54.64) | −1.61 | 45 | 0.114 | 0.47 |
| Physical activity per week (minutes) | 47 | 146.93 | (136.19) | 160.98 | (128.06) | −0.36 | 45 | 0.717 | 0.11 |
| BMI (kg/m2) | 41 | 31.69 | (4.02) | 29.39 | (4.32) | 1.74 | 39 | 0.090 | 0.55 |
| **Mental Health** | | | | | | | | | |
| Depression | 41 | 17.45 | (9.37) | 6.3 | (5.16) | 4.85 | 39 | 0.000 | 1.53 |
| State Anxiety | 41 | 43.31 | (10.20) | 34.4 | (7.26) | 3.27 | 39 | 0.002 | 1.03 |
| Trait Anxiety | 41 | 47.58 | (8.57) | 35.5 | (6.20) | 5.24 | 39 | 0.000 | 1.65 |
| Health Locus of Control | 41 | 35.51 | (5.43) | 35.6 | (4.38) | −0.06 | 39 | 0.951 | −0.02 |
| Task-oriented Coping | 41 | 47.85 | (13.89) | 60.33 | (10.88) | −3.23 | 39 | 0.003 | −1.02 |
| Emotion-oriented Coping | 41 | 39.08 | (7.86) | 35.25 | (5.90) | 1.78 | 39 | 0.082 | 0.56 |
| Avoidance-oriented Coping | 41 | 39.91 | (5.27) | 43.17 | (7.83) | −1.51 | 39 | 0.138 | −0.48 |
| Difficulty Identifying Feelings | 41 | 16.11 | (5.40) | 12.56 | (1.89) | 2.95 | 39 | 0.005 | 0.93 |
| Difficulty Describing Feelings | 41 | 11.44 | (3.78) | 10.22 | (1.34) | 1.43 | 39 | 0.160 | 0.45 |
| Externally-oriented Thinking | 41 | 22.11 | (5.98) | 18.44 | (3.73) | 2.40 | 39 | 0.021 | 0.76 |
| Alexithymia | 41 | 49.66 | (13.34) | 41.22 | (5.48) | 2.76 | 39 | 0.009 | 0.87 |
| Cognitive Reappraisal | 23 | 5.06 | (1.37) | 5.06 | (1.45) | −0.01 | 21 | 0.993 | 0.00 |
| Suppression | 23 | 4.31 | (0.78) | 3.77 | (0.81) | 1.63 | 21 | 0.118 | 0.68 |

Note: N is the number of non-missing values.

ANCOVAs of follow-up outcomes controlling for baseline confirmed the higher levels of HDL cholesterol ($F_{1,43} = 5.87$; $p = 0.020$) and the lower levels of tryglicerids ($F_{1,40} = 11.22$; $p = 0.002$) and systolic blood pressure ($F_{1,43} = 4.19$; $p = 0.047$) in the IT group. No differences in lifestyle outmes were found. Regarding psdhological outcomes, ANCOVAs higlighted better outcomes in the IT group. Both depression ($F_{1,38} = 18.75$; $p < 0.001$) and anxiety levels ($F_{1,38} = 6.25$; $p = 0.017$ and $F_{1,38} = 16.95$; $p < 0.001$ for state and trait scores, respectively) were lower in the IT group. Similarly, patients in the IT group were less alexithyic than those in the ST group at follow-up ($F_{1,38} = 6.38$, $p = 0.016$; $F_{1,38} = 8.03$, $p = 0.007$; and $F_{1,38} = 6.28$; $p < 0.017$ for total score, difficulty identifying feelings, and externally-oriented thinking, respectively). Last, ANCOVAs confirmed greater task-oriented coping ($F_{1,38} = 9.16$; $p = 0.004$) and the lower emotion-oriented coping ($F_{1,38} = 4.15$; $p = 0.049$) for the IT group at follow up.

### 3.3. Longitudinal Change

Table 4 reports the Pre-Post Change scores for the two groups separately. Consistent with cross-sectional analyses, hBa1C and blood glucose levels decreased significantly in both treatment groups. Regarding allostatic load, the total blood cholesterol level decreased significantly in both groups, but the LDL cholesterol decreased significantly in the IT group, while this effect was only marginally significant in the ST group. The HDL cholesterol significantly decreased in the ST only. At the same time, patients taking the IT maintained a relatively healthy level of HDL cholesterol. Notably, the triglycerides level of the IT group tended to decrease at follow-up compared to baseline, while the ST group significantly

increased. The Systolic Blood Pressure remained stable during the study period, while the Diastolic one significantly decreased in both groups. Regarding patients' lifestyles, both groups decreased the number of cigarettes smoked per day and increased physical activity, both daily and weekly. The IT group significantly decreased their daily wine consumption. The analysis of psychological outcomes revealed a different trajectory of the two groups during the study. Scores on the depression scale tended to increase in ST, while they tended to decrease in IT. Likewise, patients in the IT group decreased their trait–anxiety scores significantly, while those in the ST maintained their baseline level. The analyses also showed that IT patients tended to increase their Situational Coping skills and significantly reduced Emotional Coping. Similarly, patients in the integrated treatment group reduced their difficulties in identifying emotions.

**Table 4.** Change scores for IT and ST groups.

| Outcome | Treatment | Pre-Post Change | 95% Confidence Interval | $t$ | $p$-Level (Two-Tailed) |
|---|---|---|---|---|---|
| hBa1C (%) | Integrated | −1.48 | [−2.03; −0.92] | −5.37 | 0.000 |
| | Standard | −1.10 | [−1.68; −0.53] | −3.88 | 0.001 |
| Blood glucose (mg/dL) | Integrated | −58.17 | [−86.31; −30.03] | −4.16 | 0.000 |
| | Standard | −51.91 | [−81.91; −21.92] | −3.49 | 0.001 |
| Total Cholesterol (mg/dL) | Integrated | −14.41 | [−25.38; −3.44] | −2.65 | 0.011 |
| | Standard | −13.73 | [−25.42; −2.03] | −2.36 | 0.022 |
| HDL Cholesterol (mg/dL) | Integrated | −16.13 | [−16.13; −6.25] | −3.29 | 0.002 |
| | Standard | −10.55 | [−10.55; 0.16] | −1.98 | 0.053 |
| LDL Cholesterol (mg/dL) | Integrated | 3.21 | [−3.62; −3.62] | 0.95 | 0.349 |
| | Standard | −8.52 | [−15.92; −15.92] | −2.32 | 0.025 |
| Triglycerides (mg/dL) | Integrated | −13.01 | [−28.94; 2.91] | −1.65 | 0.106 |
| | Standard | 25.94 | [7.18; 44.71] | 2.79 | 0.008 |
| Systolic Blood Pressure (mmHg) | Integrated | −3.22 | [−8.57; 2.12] | −1.21 | 0.231 |
| | Standard | −1.48 | [−7.27; 4.30] | −0.52 | 0.608 |
| Diastolic Blood Pressure (mmHg) | Integrated | −4.20 | [−7.50; −0.90] | −2.56 | 0.014 |
| | Standard | −3.58 | [−7.10; −0.07] | −2.05 | 0.046 |
| Cigarettes per day (number) | Integrated | −3.88 | [−7.50; −0.73] | −2.48 | 0.017 |
| | Standard | −5.73 | [−7.10; −2.38] | −3.44 | 0.001 |
| Wine per day (glasses) | Integrated | −0.83 | [−1.31; −0.34] | −3.46 | 0.001 |
| | Standard | −0.13 | [−0.72; 0.46] | −0.46 | 0.650 |
| Alcohol per day (glasses) | Integrated | 0.79 | [−1.07; 2.66] | 0.86 | 0.395 |
| | Standard | −0.07 | [−2.51; 2.37] | −0.06 | 0.953 |
| Walk per day (minutes) | Integrated | 31.51 | [18.68; 44.33] | 4.95 | 0.000 |
| | Standard | 20.05 | [6.37; 33.72] | 2.95 | 0.005 |
| Physical activity per week (minutes) | Integrated | 109.54 | [73.12; 145.96] | 6.06 | 0.000 |
| | Standard | 77.63 | [38.71; 116.54] | 4.02 | 0.000 |
| BMI (kg/m2) | Integrated | −0.47 | [−2.01; 1.07] | −0.61 | 0.543 |
| | Standard | −0.54 | [−2.29; 1.21] | −0.63 | 0.535 |
| Depression | Integrated | −3.00 | [−6.46; 0.46] | −1.76 | 0.087 |
| | Standard | 3.31 | [−0.60; 7.22] | 1.71 | 0.095 |
| State Anxiety | Integrated | −4.18 | [−9.27; 0.91] | −1.66 | 0.104 |
| | Standard | −3.64 | [−9.39; 2.10] | −1.28 | 0.207 |
| Trait Anxiety | Integrated | −5.33 | [−9.33; −1.32] | −2.69 | 0.010 |
| | Standard | −1.44 | [−5.96; 3.09] | −0.64 | 0.525 |
| Health Locus of Control | Integrated | 1.34 | [−1.53; 3.87] | 0.88 | 0.386 |
| | Standard | 1.51 | [−5.23; 0.88] | −1.44 | 0.157 |
| Task-oriented Coping | Integrated | 6.12 | [−0.82; 13.05] | 1.78 | 0.082 |
| | Standard | −1.94 | [−9.78; 5.90] | −0.50 | 0.620 |
| Emotion-oriented Coping | Integrated | −6.14 | [−10.33; −1.95] | −2.96 | 0.005 |
| | Standard | −0.35 | [−5.09; 4.39] | −0.15 | 0.881 |
| Avoidance-oriented Coping | Integrated | 0.21 | [−4.51; 4.93] | 0.09 | 0.929 |
| | Standard | 1.20 | [−4.13; 6.53] | 0.45 | 0.652 |

| Outcome | Treatment | Pre-Post Change | 95% Confidence Interval | *t* | *p*-Level (Two-Tailed) |
|---|---|---|---|---|---|
| Difficulty Identifying Feelings | Integrated | −4.14 | [−10.38; 2.10] | −1.34 | 0.187 |
| | Standard | −1.76 | [−8.81; 5.29] | −0.51 | 0.616 |
| Difficulty Describing Feelings | Integrated | −2.67 | [−5.36; 0.01] | −2.01 | 0.051 |
| | Standard | −0.36 | [−3.39; 2.68] | −0.24 | 0.814 |
| Externally-oriented Thinking | Integrated | −1.73 | [−3.74; 0.28] | −1.74 | 0.089 |
| | Standard | −2.18 | [−4.45; 0.09] | −1.94 | 0.060 |
| Alexithymia | Integrated | −1.73 | [−3.74; 0.28] | −1.74 | 0.089 |
| | Standard | −2.18 | [−4.45; 0.09] | −1.94 | 0.060 |
| Cognitive Reappraisal | Integrated | 0.20 | [−0.54; 0.94] | 0.56 | 0.579 |
| | Standard | 0.43 | [−0.36; 1.22] | 1.13 | 0.270 |
| Suppression | Integrated | 0.05 | [−0.68; 0.77] | 0.13 | 0.897 |
| | Standard | 0.05 | [−0.73; 0.83] | 0.13 | 0.898 |

*3.4. Economic Outcomes*

Table 5 reports the number of referrals to a specialist cumulatively assessed during the study and the associated standard costs. Patients in the integrated care condition had needed fewer cardiological and diabetological visits than patients in standard care. The associated costs were also statistically significant. No differences were found in the number of ophthalmological referrals. The number of visits to the general practitioner was also significantly lower for the IT group, and, similarly, this group needed fewer blood tests or diagnostic investigations during the study. Overall, there was an average difference of one visit and an average saving of about 80 EUR between treatments.

**Table 5.** Referrals to a specialist and associated standard costs for standard and integrated treatment groups.

| | | Standard Treatment | | Integrated Treatment | | | | | |
|---|---|---|---|---|---|---|---|---|---|
| | N | M | SD | M | SD | *t*-Test | df | *p* | Cohen's d |
| Ophthalmic referrals needed | 41 | 0.94 | 0.24 | 0.83 | 0.39 | 1.14 | 39 | 0.261 | 0.36 |
| Cost of Ophthalmic examination | 41 | 60.44 | 15.08 | 52.87 | 24.8 | 1.14 | 39 | 0.261 | 0.36 |
| Cardiology referrals needed | 41 | 1.33 | 0.84 | 0.78 | 0.42 | 2.74 | 39 | 0.009 | 0.86 |
| Cost of Cardiology referrals | 41 | 85.33 | 53.77 | 50.09 | 26.99 | 2.74 | 39 | 0.009 | 0.86 |
| Diabetological referrals needed | 41 | 2.06 | 0.24 | 1.48 | 0.51 | 4.43 | 39 | <0.001 | 1.39 |
| Cost of Diabetological referrals | 41 | 131.56 | 15.08 | 94.61 | 32.69 | 4.43 | 39 | <0.001 | 1.39 |
| Total Blood tests performed | 50 | 2.63 | 0.99 | 2.1 | 0.57 | 2.34 | 48 | 0.024 | 0.66 |
| Total Diagnostic exams preformed | 50 | 4.35 | 0.85 | 3.19 | 0.79 | 4.97 | 48 | <0.001 | 1.41 |
| Visits to the general practitioner | 50 | 1.66 | 1.58 | 0.46 | 0.78 | 3.41 | 48 | 0.001 | 0.97 |
| Total Visits | 41 | 4.33 | 0.84 | 3.09 | 0.85 | 4.69 | 39 | <0.001 | 1.48 |
| Total Costs | 41 | 277.33 | 53.77 | 197.57 | 54.28 | 4.69 | 39 | <0.001 | 1.48 |

**4. Discussion**

The primary medical endpoints of the study were fasting blood glucose and glycated hemoglobin laboratory assays. Both variables decreased significantly over six months regardless of the treatment. This finding did not support the superiority of the IT over standard care [8]. Allostatic overload is increasingly recognized as the interface between body functioning and these "environmental" factors, accounting for phenotype variability in diabetes variability in diabetes [29]. Our study, therefore, tested whether the IT reduced patients' allostatic load using blood lipids as primary markers. While the two treatments were not statistically different at baseline, the IT group had significantly lower blood lipids and triglyceride levels at follow-up than the ST. No difference was found in blood pressure.

These findings suggested the advantage of IT over standard care, subsequently reinforced by longitudinal change analysis. While total cholesterol decreased in both treatments,

only the IT healthy levels of "good" cholesterol (HDL) were preserved, together with a drastic drop in triglycerides and the relative stability of "bad" cholesterol (LDL). There is no evidence that dysregulation of blood lipids can directly affect glycemia; in fact, blood sugar levels and dyslipidemia can result from insulin resistance [58]. However, elevated LDL and triglycerides may increase the risk of complications in T2DM, such as peripheral neuropathy or diabetic retinopathy, to cite a few [59,60]. In summary, our results showed that integrated care, although it did not add a specific contribution in controlling glycemia, can help control associated dyslipidemia, thus helping patients reduce the risk of the complications mentioned above.

Unexpectedly, both treatments changed lifestyle by increasing physical activity levels and reducing patients' smoking habits. Two interpretations are possible: First, the psychological intervention aimed primarily to address emotional dimensions (diabetes distress, helplessness, anxiety states) and relational ones (relationships with caregivers and health care providers). Second, ST already included health education and carbohydrate counting. The potential to influence lifestyle variables was indeed present in both groups.

The two groups started the study with similar levels of depression, the primary psychological endpoint of the study. After six months, the groups diverged in the opposite direction: patients in the integrated intervention group decreased their depression while those undergoing standard intervention tended to increase it. Likewise, the integrated intervention effectively relieved trait anxiety, a dispositional tendency to respond anxiously to many stressors [51]. Numerous studies show that depressed and anxious patients with T2DM have worse glycemic control [20,21,61], lower treatment adherence, [23], long-term complications, and higher healthcare costs [24,25]. Consistent with this literature, our study suggests that integrated care can decrease chronic depressive and anxious mood states associated with disease management in the medium to long term.

The IT also decreased the patient's emotional coping (e.g., ruminating, worrying, or venting emotions) and increased the task-oriented one (e.g., cognitively restructuring the distressing situation). The emotion-oriented coping was found to be related to negative diabetes appraisals [62,63] and accounted for unhealthy eating in response to diabetes-related stress [64]. Conversely, higher task-related coping was associated with decreased glycated hemoglobin dosage [65]. In line with this literature, our study suggests that dealing with psychological aspects may facilitate adjustment processes by activating more positive appraisals of the disease and greater use of cognitive strategies.

Last, our study provided evidence that patients receiving the IT needed fewer specialist visits, used general practitioner advice less frequently, and required fewer blood tests and diagnostic examinations than those receiving the standard care. If confirmed in future research, these findings might have implications for the sustainability of healthcare costs.

The study has some noteworthy limitations. First, for organizational constraints, it was not possible to randomize patients into the two conditions. Although the groups were similar at baseline, there were also some differences in the study variables, of which we have not assessed the impact because of the limited sample size. Second, the sample taking the IT is self-selected, and therefore may be more likely to benefit from group psychotherapy than the general population of patients with T2DM. Third, there might be some bias risk because the study was not blinded to patients, clinicians, and researchers. Last, we must acknowledge the lack of long-term data. The present study piloted the feasibility of an integrated care model for T2DM patients. Long-term follow-ups are needed for definite conclusions about its enduring effectiveness.

Before concluding, it is worth discussing some clinical implications of the therapeutic model adopted in the present study. The ability of ST to activate changes in lifestyles supported the importance and effectiveness of health education interventions. However, IT increased the sustainability of this change because it activated significant changes in salutogenic behaviors and improved the emotional states (i.e., anxiety and depression) that affect the individual's ability to mobilize energy to maintain the changes initiated over time. We believe that recognizing one's own emotions, focusing on awareness of the behaviors

they triggered, exploring and sharing one's illness experience and coping strategies, and the support and containment provided by the group favored mood stabilization made the illness threat more manageable.

In our clinical experience, the Gestalt and behavioral models proved to be advantageously integrated, showing similarities and synergies and precise specificities. The cognitive approach can be very effective in focusing on problematic and maladaptive behaviors and in deconstructing and making more manageable the challenges posed by the disease, while it has shown limits in its intervening on affective variables. In a complementary way, the Gestalt model has offered flexible and powerful tools to explore, share, and modify patients' relational and emotional worlds. Addressing chronic disease is complex and poses recurrent difficulties to the individual. For these reasons, it requires adequate and prolonged emotional and behavioral management.

## 5. Conclusions

Notwithstanding limitations, our findings provided preliminary evidence that a close collaboration between diabetologists and psychologists is feasible where this approach is not yet frequently used in adult populations with T2DM. Diabetes is a financial burden on the NHS, primarily due to complications, requiring hospitalizations, specialist visits, and diagnostic tests. Considering the expected increase in the prevalence of T2DM in Western and industrialized countries, it is worth considering integrated care as an option to contain and make healthcare spending more sustainable.

**Author Contributions:** Conceptualization, M.L. (Mara Lastretti) and M.T.; methodology, M.L. (Marco Lauriola), M.L. (Mara Lastretti); software, M.L. (Marco Lauriola); formal analysis, M.L. (Marco Lauriola); investigation, M.L. (Mara Lastretti) and M.L. (Marco Lauriola); resources, M.L. (Mara Lastretti); writing—original draft preparation, M.T. and M.L. (Marco Lauriola); writing—review and editing, M.T.; supervision, N.V., F.C. and R.T.; funding acquisition, M.L. (Mara Lastretti). All authors have read and agreed to the published version of the manuscript.

**Funding:** This research was funded by ORDINE DEGLI PSICOLOGI DEL LAZIO [grant number 336-16].

**Institutional Review Board Statement:** The study was conducted according to the guidelines of the Declaration of Helsinki, and approved by Ethics Committee of Department of Clinical Dynamic Psychology and Health Study (Sapienza-Roma) (protocol code 0000367 UOR:SI000092-Classif. VII/15 and date of approval: 02/04/2021).

**Informed Consent Statement:** Informed consent was obtained from all subjects involved in the study. Written informed consent has been obtained from the patient(s) to publish this paper.

**Data Availability Statement:** The data supporting this study's findings are available at this link: https://osf.io/hnmek/.

**Acknowledgments:** The authors are grateful to Maria Casagrande for advice related to the experimental design and selection of measures. We also express gratitude to Martina Ruggiero for collecting and organizing references concerning this research. Special thanks to Rossella Valotta for supporting the integrated treatment group and the psychotherapist. Last, we thank the healthcare team involved in the study and all patients for participation.

**Conflicts of Interest:** The authors declared no potential conflicts of interest concerning this research, authorship, and/or publication of this article.

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
