# Peer review of "An Integrated Medical-Psychological Approach in the Routine Care of Patients with Type 2 Diabetes: A Pilot Study to Explore the Clinical and Economic Sustainability of the Healthcare Intervention"

_sustainability, doi:10.3390/su132313182_

Round 1

Reviewer 1 Report

It is an interesting study for both theoretical and practical reasons. It provided evidence that collaboration between medics and psychologists is highly needed in the treatment of some diseases, especially when psychological factors are involved.  Putting aside obvious limitations which are mentioned rightly by the authors, the study was well designed and sufficiently described in the manuscript. However, I have a few comments which might help to improve the readability of the paper and its scientific soundness.

  1. The Method section.  Descriptive statistics in Table 1 would be more informative by adding the following information:  range. i.e. min-max, and units of measure, e.g. for physical activity - minutes, hours, for drinking - glasses.
  2. Description of psychological intervention in the management of T2DM which I suppose is the key variable is quite limited [139-149]. It needs more specific information about applied methods; using only labels such as "Gestalt therapy", "Cognitive Behavioral Therapy" is not enough.
  3.  The Discussion section. In my opinion, to make integrated treatment more applicable, it would be useful to match some outcomes to specific methods of psychological intervention. I wonder whether it is possible to indicate which methods worked best, or which were useless. It should be discussed thoughtfully by the authors. 

Author Response

Q1. The Method section.  Descriptive statistics in Table 1 would be more informative by adding the following information:  range. i.e. min-max, and units of measure, e.g. for physical activity - minutes, hours, for drinking - glasses.

A1. The required information have been added to Table1.

Q2. Description of psychological intervention in the management of T2DM which I suppose is the key variable is quite limited [139-149]. It needs more specific information about applied methods; using only labels such as "Gestalt therapy", "Cognitive Behavioral Therapy" is not enough.

A2. We enriched the description of psychological intervention by providing more specific information on the methods we applied and the main techniques borrowed from the two theoretical models of reference.

Q3. The Discussion section. In my opinion, to make integrated treatment more applicable, it would be useful to match some outcomes to specific methods of psychological intervention. I wonder whether it is possible to indicate which methods worked best, or which were useless. It should be discussed thoughtfully by the authors. 

A3. Therapeutic treatment, and group treatment as well, is a process that occurs within a complex system, making it impossible to match a single outcome to a specific method. However, as you suggested, we have indicated which  psychological interventions we believe have had the greatest impact on the main outcomes. We also indicated the contribution, in terms of objectives, methods, and techniques, that we think the two therapeutic approaches have provided to the treatment model we have tested. Last, we have pointed out for which objectives Gestalt therapy and Cognitive Behavioral Therapy could be more effective.

Reviewer 2 Report

No further comments

Author Response

Thank you.

Reviewer 3 Report

This is a quasi-experimental study discussing about whether there was benefit in physical health, lifestyle, mental health, and costs after integrated treatment. I think the topic is important and contributive to patients with T2DM with an empirical approach quite valuable for clinical practice professionals.

Major concerns:

  1. Based on my knowledge, the research is only for the same period and lack of long-term data, it is difficult to investigate more deeply trends of problems or phenomena. The authors should provide some solutions to these concerns.

  1. Figure 1 always presented flow chart of this study. The authors should clarify this concern.

  1. Please use power analysis to statement adequate sample size in this study.

  1. Why didn’t use ANCOVA to address different baseline in Table 2? I suggest conducting ANCOVA for comparison of patient’s relevant outcome.

  1. Some references are outdated and should be updated accordingly.

Author Response

Q1. Based on my knowledge, the research is only for the same period and lack of long-term data, it is difficult to investigate more deeply trends of problems or phenomena. The authors should provide some solutions to these concerns.

A2. We agree that the study lacks of long-term data. The present study piloted the feasibility of an integrated care model for T2DM patients. Long-term follow-ups are needed for definite conclusions about its enduring effectiveness. We acknowledged this in the limitations of research findings.

Q2. Figure 1 always presented flow chart of this study. The authors should clarify this concern.

A2. We have now included a Figure that portrays the flow chart of the study.

Q3. Please use power analysis to statement adequate sample size in this study

A3. Thanks for this comment.  Psychosocial interventions are effective in reducing anxiety and depression symptoms in T2DM patients with a very large effect size (i.e., SMD = 1.50) and can improve glycemic control, with SMD-s ranging from 0.81 to 1.15 for fasting blood-glucose and glycated hemoglobin, respectively (Xie & Deng, 2017). We conducted an a priori power analysis to determine the sample size needed for our study using the "pwr" package for R (Champely et al., 2015).  Assuming the effect sizes mentioned above as our best guess for the expected differences between IT and ST at follow-up, a sample of 25 participants in each treatment group would detect a difference in fasting blood glucose with alpha level .05 and power .80. For glycated hemoglobin, a sample of 13 participants in each treatment group would be needed. Given the higher effect size for mental health outcomes, the same analysis recommended a sample size of 8 patients in each group. A sample size of 25, would detect a significant difference in anxiety and depression, with a power of .999.

Q4. Why didn’t use ANCOVA to address different baseline in Table 2? I suggest conducting ANCOVA for comparison of patient’s relevant outcome.

A4. As per your recommendation we carried out an ANCOVA to address different baselines in Table 2.  The results are reported in section 3.2 "Patients Characteristics at Follow up" just after t-tests. No ostensible differences between ANCOVAs and t-tests were observed. ANCOVAs confirmed previously reported findings.

Q5. Some references are outdated and should be updated accordingly.

A5. Thank you very much. The references have been updated.

Round 2

Reviewer 3 Report

Thanks for your great efforts on revision. Only a minor comment, I suggest that legend of Figure 1 should be belowly inserted of flow chart.